# A Novel Density Peaks Clustering Algorithm with Isolation Kernel and K-Induction

Shichen Zhang [1] and Kai Li [1,2,*]

[1] School of Cyber Security and Computer, Hebei University, Baoding 071002, China
[2] Hebei Machine Vision Engineering Research Center, Hebei University, Baoding 071000, China
[*] Correspondence: likai@hbu.edu.cn

**Abstract:** Density peaks clustering (DPC) algorithm can process data of any shape and is simple and intuitive. However, the distance between any two high-dimensional points tends to be consistent, which makes it difficult to distinguish the density peaks and easily produces "bad label" delivery. To surmount the above-mentioned defects, this paper put forward a novel density peaks clustering algorithm with isolation kernel and K-induction (IKDC). The IKDC uses an optimized isolation kernel instead of the traditional distance. The optimized isolation kernel solves the problem of converging the distance between the high-dimensional samples by increasing the similarity of two samples in a sparse domain and decreasing the similarity of two samples in a dense domain. In addition, the IKDC introduces three-way clustering, uses core domains to represent dense regions of clusters, and uses boundary domains to represent sparse regions of clusters, where points in the boundary domains may belong to one or more clusters. At the same time as determining the core domains, the improved KNN and average similarity are proposed to assign as many as possible to the core domains. The K-induction is proposed to assign the leftover points to the boundary domain of the optimal cluster. To confirm the practicability and validity of IKDC, we test on 10 synthetic and 8 real datasets. The comparison with other algorithms showed that the IKDC was superior to other algorithms in multiple clustering indicators.

**Keywords:** DPC; high-dimensional data; three-way clustering; isolation kernel; K-induction





## 1. Introduction

As the volume and dimensionality of data continue to increase, the focus has moved to how to analyze and process this data effectively. As a powerful data analysis tool [1,2], clustering has been widely applied in pattern recognition [3,4], intrusion detection [5,6], medicine [7,8], sensors [9,10], and many other fields.

Clustering can generally be categorized by partition [11,12], hierarchy [13,14], density [15–18], grid [19,20], model [21], and graph [22]. k-means [11] is a partition-based clustering algorithm. Although this algorithm can handle complex data well and has low complexity, it has many disadvantages; for example, it depends heavily on the input parameter *k* and is sensitive to outliers. To find the optimal parameter *k*, Dinh et al. [23] used the silhouette coefficient to evaluate the quality of different clusters to find the optimal solution. Density-based clustering can also avoid the above problems well. It finds the cluster center and noise by calculating the density of each sample point and avoids the influence of setting the number of clusters and noise in advance on the results. Some of the more classic ones are density-based spatial clustering of applications with noise (DBSCAN) [24] and DPC [25].

DBSCAN [24] can discover arbitrarily shaped clusters in the noise space with just two parameters, making the clustering results almost independent of the traversal order of the nodes. However, DBSCAN strongly depends on the distance and is not suitable for datasets with small density differences. However, DBSCAN has a strong dependence on the

parameter and is not suitable for datasets with small differences in density. DBSCAN is only effective for spherical data and has a poor impact on aspheric data. Campello et al. [26] proposed the hierarchical density-based spatial clustering of applications with noise (HDB-SCAN). It combined DBSCAN with hierarchical clustering, redefined the distance between samples, and solved the problem that DBSCAN relies heavily on input parameters. However, this algorithm does not work well with boundary points.

DPC was proposed by Alex et al. [25]. It eliminates the drawbacks of DBSCAN and HDBSCAN, which are only valid for spherical data, can find density peak points with fewer parameters, and can efficiently assign points and remove noise. Meanwhile, DPC can automatically find the cluster centers and realize efficient clustering of data of any shape. However, there are still some drawbacks with DPC, mainly in two aspects: (1) poor applicability in high-dimensional spaces. With the advent of massive data, the dimensionality of the data is growing exponentially. When the dimensionality of data is high, it can be proven by the theorem of large numbers that the sum of independent random variables converges to a fixed value when the size of the variables is large, at which time the distances between high-dimensional samples tend to be equal. This means that the traditional distance formula becomes meaningless when calculating the distance between high-dimensional samples, which leads to a poor clustering effect; (2) error message delivery. The leftover points are directly allocated to the cluster in which the closest and denser point is located, without consideration of the information of the neighboring points. If a point is incorrectly assigned during this process, this error will be passed on to subsequent points, causing points that should not belong to a cluster to be assigned to a cluster, thus resulting in clustering failure; (3) deviation from human semantic interpretation. DPC belongs to hard clustering and does not take human semantic interpretation into account.

Aiming at the poor practicability of DPC in high-dimensional data, Xu et al. [27] introduced the grid granularity framework into DPC to address the poor clustering of DPC on large-scale datasets. Du et al. [28] introduced Principal Component Analysis (PCA) into the KNN-DPC model to perform dimensionality reduction operations on high-dimensional data before clustering. Although the above two algorithms have good feasibility and can obtain satisfactory results on relatively high-dimensional datasets, when the dataset presents vertical stripes, the clustering can be reduced to poor results. Hu et al. [29] used PCA, autoencoder (AE), and t-distributed stochastic neighbor embedding (t-SNE) to automatically extract features from high-dimensional data and then used Quasi-Monte Carlo (QMC) to estimate the distribution of low-dimensional samples to obtain clustering results. Mohamed et al. [30] first convert high-dimensional data into a two-dimensional space using t-SNE and cluster low-dimensional data using mutual nearest neighbors. The above two algorithms both use dimensionality reduction technology to process high-dimensional data, which can easily cause information loss. Wang et al. [31] provided feasible channels for data of different dimensions to achieve dimensionality reduction of complex data.

Aiming at the "bad label" delivery in the DPC, Yu et al. [32] improved the validity of DPC by collecting information on the dynamics of nearest neighbors. Li et al. [33] put all data points into polar coordinates, calculated the distance and cosine values between any two points, and finally output the results with a sparse matrix. It can be efficient to minimize the running time of DPC. Liu et al. [34] proposed a two-step clustering using shared nearest-neighbor information. Sun et al. [35] redefined local density by taking into account information about nearest neighbors. Chen et al. [36] grouped samples with k of the same nearest neighbors into a cluster. Seyedi et al. [37] established a similarity fusion matrix to dynamically update the label information. All the above algorithms consider only the K nearest neighbors around the leftover points and lack consideration of global information. Yu et al. [38] proposed the use of evidence theory to collect and fuse neighbor information of the leftover samples, which can effectively solve the error message delivery. However, it is difficult to solve the conflict problem, and the fusion accuracy is not enough.

To surmount the above-mentioned defects, this paper put forward a novel density peaks clustering algorithm with isolation kernel and K-induction. The following are the major contributions:

1.  The isolation kernel is introduced and optimized. In this paper, the traditional distance is replaced by an optimized isolation kernel, which increases the similarity of two samples in a sparse domain and decreases the similarity of two samples in a dense domain. It solves the problem of poor applicability of high-dimensional data, further improves the accuracy of the algorithm, and solves the failure of traditional distance in high-dimensional spaces;

2.  Three-way clustering is introduced, and the interval sets of the core domain and boundary domain are used to represent the clustering result. First, the KNN based on the optimized isolation kernel is exploited to find the *K* neighbors of the core domain center, and the initial core domain is obtained. Then, the range of the initial core domain is continuously expanded through the average similarity to obtain the final core domain;

3.  The K-induction similarity is proposed. According to the K-induction similarity, each leftover point is allocated to the boundary domain of the cluster where the neighbor point with the largest K-induction similarity is located, which effectively avoids the domino effect and better solves the "bad label" delivery.

The rest of the paper is as follows: The DPC, the isolation kernel, and the three-way clustering are detailed in Section 2. The IKDC is further elaborated upon in Section 3. The preliminary results are displayed in Section 4 to support the efficiency of the IKDC. The implications and future work are mentioned in Section 5.

## 2. Preliminaries

The main symbols used in this section are listed in Table 1.

**Table 1.** Symbols and descriptions.

| Symbol | Description |
| --- | --- |
| $X^D$ | global sample points |
| $X'$ | a subset of sample points in the universe |
| $x_i$ | *i*-th sample point |
| $\rho_i$ | local density of sample point *i* |
| $\delta_i$ | distance from $x_i$ to the closest point with higher density than it |
| $\theta$ | isolation partition |
| $C_r^P$ | the core domain of the *r*-th cluster |
| $C_r^B$ | the boundary domain of the *r*-th cluster |
| $C_r^N$ | the noise domain of the *r*-th cluster |

### 2.1. Density Peaks Clustering

The DPC finds the peak density points by calculating the local density $\rho_i$. The DPC uses them as cluster centers, and then the leftover points are allocated directly to the cluster where the denser and closest points are located. For any point $x_i$, the local density of $x_i$ is $\rho_i$, as shown in Equation (1).

$$\rho_i = \sum_{j=1}^{n} \chi(d_{ij} - d_c) \tag{1}$$

where $n$ is the number of points, $d_{ij}$ is the distance between $x_i$ and $x_j$, $d_c$ is the cutoff distance, and $\chi(x)$ is the logical function when $x < 0$, $\chi(x) = 1$, otherwise $\chi(x) = 0$.

$\delta_i$ is the shortest distance between $x_i$ and a point denser than it, as shown in Equation (2). If no point with a higher density is found, the local density is made to be the farthest distance in the dataset.

$$\delta_i = \min_{i: \rho_i < \rho_j} (d_{ij}) \tag{2}$$

According to the above definition, by constructing a decision diagram of $\delta_i$ relative to $\rho_i$, the points are classified into three different types, i.e., density peaks, normal points, and noise. The density peaks are chosen as the cluster centers, and then the DPC allocates the leftover points to denser and the closest clusters.

### 2.2. Isolation Kernel

The isolation kernel is a special kernel function that uses the data isolation mechanism to design a similarity measurement method. It can increase the similarity between points in low-density areas and reduce the similarity between points in high-density areas. Different from traditional distances or kernels, isolation kernel [39,40] has no closed expressions and is stemmed directly from a dataset without learning [41]. The similarity between the two points is measured based on the partitions created in the data space.

A key requirement for an isolation kernel is a spatial partitioning mechanism that isolates a point from other points in the sample set. There are many kinds of spatial segmentation mechanisms, the commonly used ones are isolation forest [42,43], Voronoi Diagram [44,45] and isolating hyperspheres [46,47]. The definition of isolation kernel is as follows [39,40]: Suppose $\mathbb{H}_\psi(X^D)$ represents the set of all admissible partitions of $H$ derived from a given finite dataset $X^D$, where each $H$ is from a random subset $X' \subset X^D$, and each point in $X'$ has an equal probability of being chosen from $X^D$, and $|X'| = \psi$. The role of the isolation partition $\theta(\theta \in H)$ is to isolate the point $z \in X'$ from other points in $X'$ in a given random subset $X'$. The union of $\theta$ covers the entire dataset.

Given a data space $X^D$, the isolation kernel about $X^D$ between any two points, $x, y \in X^D$ is defined as the density distribution expectation that all $H$, $x$, and $y$ belong to the same isolation partition $\theta$, as shown in Equation (3).

$$K_\psi\left(x, y \mid X^D\right) = \mathbb{E}_{\mathbb{H}_\psi(X^D)}[\mathbb{I}(x, y \in \theta \mid \theta \in H)] \tag{3}$$

where $\mathbb{I}(B)$ is the indicator function; if $B$ is true, it outputs 1, otherwise, it outputs 0.

In fact, the isolation kernel $K_\psi$ is computed through a partition $H_m \in \mathbb{H}_\psi(X^D)$ $(m = 1, 2, \cdots, t)$, as shown in Equation (4).

$$K_\psi(x, y | X^D) = \frac{1}{t} \sum_{m=1}^{t} \sum_{\theta \in H_{mi}} \mathbb{I}(x \in \theta)\mathbb{I}(y \in \theta) \tag{4}$$

where $K_\psi(x, y) \in [0, 1]$.

### 2.3. Three-Way Clustering

Given a data space $X^D = \{x_1, \cdots, x_i, \cdots, x_n\}$, where $x_i = \{x_i^1, x_i^2, \cdots, x_i^D\}$, each object $x_i$ has D-dimensional attributes. The traditional cluster result is shown as $C = \{C_1, \cdots, C_r, \cdots, C_k\}$, where $k$ is the number of clusters, which does not conform to the semantic interpretation of human thinking. Three-way clustering (3w) [48–50] introduces the idea of a three-way decision [51,52] and divides the global data space into $C_k^P$, $C_k^B$, and $C_k^N$, which are, respectively, the core domain, boundary domain, and noise domain of the cluster, as shown in Equation (5). The samples in the core domains definitely belong to this cluster, the samples in the boundary domains may or may not belong to this cluster, and the samples in the noise domain definitely do not belong to this cluster. Three-way clustering uses two disjoint sets ($C_r^P \cup C_r^B$) to represent a cluster, which is in accordance with human cognition.

$$C_r^P \cup C_r^B \cup C_r^N = X^D \tag{5}$$

## 3. The Model of IKDC

As the dimensionality of the data becomes higher and higher, the distances between points tend to be equal, and the traditional Euclidean distance metric fails in high-dimensional space. In addition, DPC will assign the leftover points to the closest and denser clusters. If a point is incorrectly assigned, this "bad label" will be passed to the next leftover point that is assigned, which will have a domino effect. To surmount the above-mentioned defects, this paper will make improvements to the DPC in two directions: (1) introducing a new distance based on isolation kernel; (2) defining the K-induction similarity to assign the leftover points and assigning them to the boundary domain of the cluster with the highest K-induced similarity. In this section, the basic details of the IKDC will be given, and its time complexity will be theoretically analyzed.

### 3.1. Determining Cluster Centers and Noise Points

When the dimensionality of data is high, it can be proved by the theorem of large numbers that the sum of independent random variables converges to a fixed value when the size of the variables is large, and the distances between any high-dimensional samples tend to be equal. At this time, density peak points cannot be found using the traditional distance formula, resulting in poor clustering results. Therefore, to solve the above problem, this paper introduces the isolation kernel and improves it. The isolation kernel can increase the similarity of samples in the sparse domain and reduce the similarity of samples in the dense domain, effectively solving the problem of consistent similarity between high-dimensional samples. From this, it can be seen that compared with the distance calculation metric used by the DPC, the isolation kernel is more suitable for calculating the distance between high-dimensional samples. In this paper, the space segmentation mechanism of the isolation forest is used to construct the isolation kernel, and function $K_\psi$ is used to replace the distance calculation metric in the DPC. The optimizing distance based on the isolation kernel is further obtained, as shown in Equation (6).

**Definition 1** (distance based on isolation kernel). *Given a data space $X^D$, the distance $\tilde{d}_{ij}$ based on the isolation kernel between points $x_i$ and $x_j$ is given by Equation (6).*

$$\tilde{d}_{ij} = \frac{1}{K_\psi(x_i, x_j | D) + \frac{1}{t}} \tag{6}$$

*where t is the number of total trees in the isolation forest.*

IKDC uses the distance based on the isolation kernel instead of the traditional distance and then follows the idea of DPC to find cluster centers and noise. The cluster centers and noise are put into the core domains and noise domain $\Omega$, respectively, and the noise domain is removed from the global dataset to put the effect of noise on the clustering results to the minimum.

To overcome the problem that the KNN algorithm fails in high-dimensional space, the improved KNN is obtained by using the distance based on the isolation kernel instead of the distance formula in the KNN algorithm.

### 3.2. Using Improved KNN and the Average Distance to Determine the Core Domains

After determining the cluster centers and noise points, IKDC first removes the noise points from $X^D$, uses the improved KNN algorithm to determine the initial core domains in the set $X^D - \Omega$, denoted as $q_j (j \in [1, K])$, and then continuously expands the range of the core domain through the average distance to obtain the final core domain.

**Definition 2** (the *K*-nearest neighbors). *Given a data space $X^D$, for any point p in the dataset, its K-nearest neighbor representation is shown in Equation (7).*

$$K_p = \{q_j | j \in [1, K]\} \tag{7}$$

*where K is the number of nearest neighbors.*

**Definition 3** (the average similarity). *Given a data space $X^D$ and the point p, the average similarity is shown in Equation (8).*

$$d_p^{avg} = \frac{1}{K} \sum_1^K \tilde{d}_{pq_j} \tag{8}$$

*where $\tilde{d}_{pq_j(j \in [1,K])}$ is the distance based on the isolation kernel between point p and its j-th nearest neighbor.*

Based on the initial core domain, the final core domain is obtained by continuously expanding the core domain range through the average similarity, as shown in Equation (8). More details are shown in Algorithm 1.

---

**Algorithm 1** Determining the core domain based on the average similarity

---

Input: cluster center $c = \{c_1, \cdots, c_r, \cdots, c_k\}$, data space $X^D - \Omega$
Output: core domain $\{c_1^p, c_2^p, \cdots, c_k^p\}$

  1: **for** $r = 1$, $r <= k$ **do**
  2:     find *K*-nearest neighbors of $c_r \rightarrow KNN_{c_r}$;
  3:     $C_r^P = [c_r; KNN_{c_r}]$;
  4:     initialize $Q = KNN_{c_r}$;
  5:     **while** $Q$ is not empty **do**
  6:        select the head point $p$;
  7:        find its *K*-nearest neighbors $KNN_p = \{q_1, \cdots, q_j, \cdots, q_K\}$;
  8:        $d_p^{avg} = \frac{1}{k} \sum_1^K d_{pq_j}$;
  9:        **for** each point $q_j$ in $KNN_p$ **do**
10:           **if** $d_{pq_j} <= d_p^{avg}$ **then**
11:              $C_r^P = C_r^P \cup q_j$;
12:              add $q_j$ to the end of $Q$;
13:           **end if**
14:        **end for**
15:        remove $p$ from the head of $Q$;
16:     **end while**
17: **end for**
18: **return** $\{c_1^p, c_2^p, \cdots, c_k^p\}$

---

*3.3. Using K-Induction to Assign Remaining Points*

After the final core domains of the cluster are determined by Algorithm 1, K-induction is used to assign the other remaining points to the boundary region of the determined cluster. The main idea is that, for a point *p*, the *K* neighbors of its surrounding can be regarded as *K* pieces of information, and by fusing this information, *p* is derived as to which cluster it belongs to. The K-induction expands the search range by integrating neighbor information of its neighbor, which effectively avoids the transfer of bad labels. It can not only avoid the limitation caused by only considering neighbor information, but also reduce the time consumed when searching for global information. Therefore, this paper first calculates the K-induction between the leftover point *p* and the surrounding *K*-neighbor and, secondly allocates the leftover points to the cluster where the point with

the maximum K-induction is located. At this time, two situations may occur: (1) If there is only one neighbor $q_j$ with the maximum K-induction, it allocates the point $p$ to the boundary domains of the cluster where the neighbor with the largest K-induction is located. If $q_j$ has not been assigned, it continues to iterate until the most suitable cluster is found; (2) If there are two or more neighbor points with the maximum K-induction, then it assigns the points to the boundary domains of the clusters where these neighbor points are located. If these neighbor points have not been assigned, then it continues to iterate. More details are shown in Algorithm 2.

---

**Algorithm 2** Assign the remaining points to the boundary domain based on K-induction

---

Input: the set of R remaining points $X^D - \Omega - \left\{ C_1^P, C_2^P, \cdots, C_k^P \right\}$
Output: boundary domain $\left\{ C_1^B, C_2^B, \cdots, C_k^B \right\}$

1: **for** each remaining point $p_i$ **do**
2:     calculate the neighbor similarity between point $p_i$ and the $j$-th $K$-nearest-neighbor in the direction of $q_j$ using Equations (9)–(12);
3: **end for**
4: construct a neighbor similarity matrix $S = \left[ \text{s}(p_i, q_j) \right]_{R \times K}$;
5: **while** $\exists$ the maximum $s = (p_i, q_j)$ in $S$ **do**
6:     $m, n \leftarrow \arg \max\limits_{i \in [1.R], j \in [1,K]} s(p_i, q_j)$;
7:     $C_n^B = C_n^B \cap p_m$;
8:     **for** each unassigned point $e$ satisfying $p_m \underset{e, p_m}{\in} KNN_e$ **do**
9:         calculate the neighbor similarity between point $e$ and the $m$-th $K$-nearest-neighbor in the direction of $q_m$ using Equations (9)–(12);
10:         update the related values in the matrix $S$;
11:     **end for**
12: **end while**
13: **for** each fringe point $p_i$ **do**
14:     calculate the neighbor similarity between point $p_i$ and the $j$-th $K$-nearest-neighbor in the direction of $q_j$ using Equations (9)–(12);
15:     $n \leftarrow \arg \max\limits_{i \in [1.R], j \in [1,K]} s(p_i, q_j)$;
16:     $C_n^B = C_n^B \cap p_i$;
17:     **if** the point $p$ is still not assigned **then**
18:         find the assigned point $q$ which is nearest to $p$;
19:         assign $p$ to the boundary domains of the same cluster as $q$ is located;
20:     **end if**
21: **end for**
22: **return** $\left\{ c_1^p, c_2^p, \cdots, c_k^p \right\}$

---

**Definition 4** (the neighbor difference). *Given a data space $X^D$ and the point $p$, the neighbor difference between the K-neighbors of the point $p$ and the $j$-th K-nearest neighbor $q_j$ is shown in Equation (9).*

$$\text{dif}_p(q_j) = \frac{1}{K} \sum \left\| K_p - K_{q_j} \right\| \tag{9}$$

**Definition 5** (the neighbor weight). *Given a data space $X^D$ and the point $p$, the weight of the point $p$ and the $j$-th K-nearest neighbor $q_j$ is shown in Equation (10).*

$$\omega_p(q_j) = \frac{\exp\left(-\alpha dif_p(q_j)\right)}{\sum_{j \in [1,K]} \exp\left(-\alpha dif_p(q_j)\right)} \tag{10}$$

*where $\alpha$ is an input parameter in $(0, 1)$.*

**Definition 6** (the weighted difference). *Given a data space $X^D$ and the point p, the weighted difference between the point p and the j-th K-nearest neighbor in the direction of $q_j$ is shown in Equation (11).*

$$d_p^\omega(q_j) = \frac{1}{K}\left(\omega_p(q_j) \otimes \left\| K_p - K_{q_j} \right\|\right) \tag{11}$$

*where $\otimes$ represents the dot product of two vectors.*

**Definition 7** (the K-induction similarity). *Given a data space $X^D$ and the point p, the K-induction similarity of point p with the j-th K-nearest neighbor $q_j$ is shown in Equation (12).*

$$s(p, q_j) = \exp\left(-d_p^\omega(q_j)\right) \tag{12}$$

Given the original data distribution and core domain range as shown in Figure 1, the detailed steps for calculating the K-induction similarity of unassigned points are shown in Figure 2, where $\alpha = 0.05$, $K = 4$.

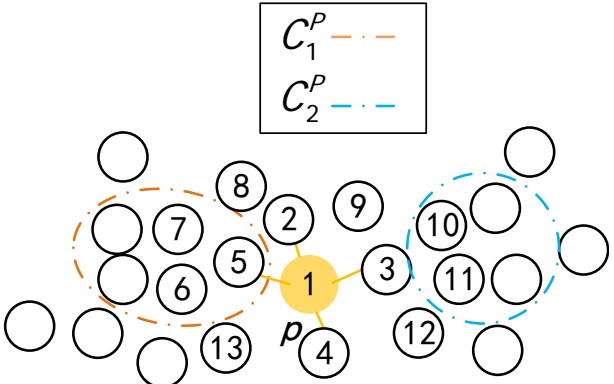

**Figure 1.** Raw data distribution and core domain range.

The K-induction similarity of unassigned point 1 with its four neighbors (2, 3, 4, 5) and the K-induction similarity of neighboring points 2, 3, 4 and 5 with their neighbors are shown in Figure 2. The neighbor difference between unassigned points 1 and 2, 3, 4 and 5 is obtained by Equation (9), and based on this, the neighbor weight is calculated using Equation (10), as shown in Figure 2. Then, the weighted difference of unassigned point 1 from its neighboring points is obtained using Equation (11), as in Figure 2. Finally, the K-induction similarity between unassigned point 1 and its neighboring points is calculated using Equation (12), as in Figure 2.

### 3.4. The Computation Complexity Analysis of IKDC

The computation complexity of the IKDC is discussed below. The major difference between the IKDC and the DPC lies in Algorithms 1 and 2. Therefore, this paper limits discussion to these two algorithms. In Algorithm 1, for each data point, the distance based on the isolation kernel $\tilde{d}_{ij}$ is calculated, and the core centers are determined, resulting in O $(n^2)$ complexity. The *K*-nearest neighbors for each core center are found, resulting in O$(Kn)$ complexity. Algorithm 2 involves finding *K*-nearest neighbors and calculating K-induction for each of the remaining points. Noticing that *K*-nearest neighbors have been obtained in Algorithm 1, this algorithm has a computation complexity of O$(Kn)$. In summary, due to *K* being much smaller than *n*, IKDC has a computation complexity of O $(n^2)$.

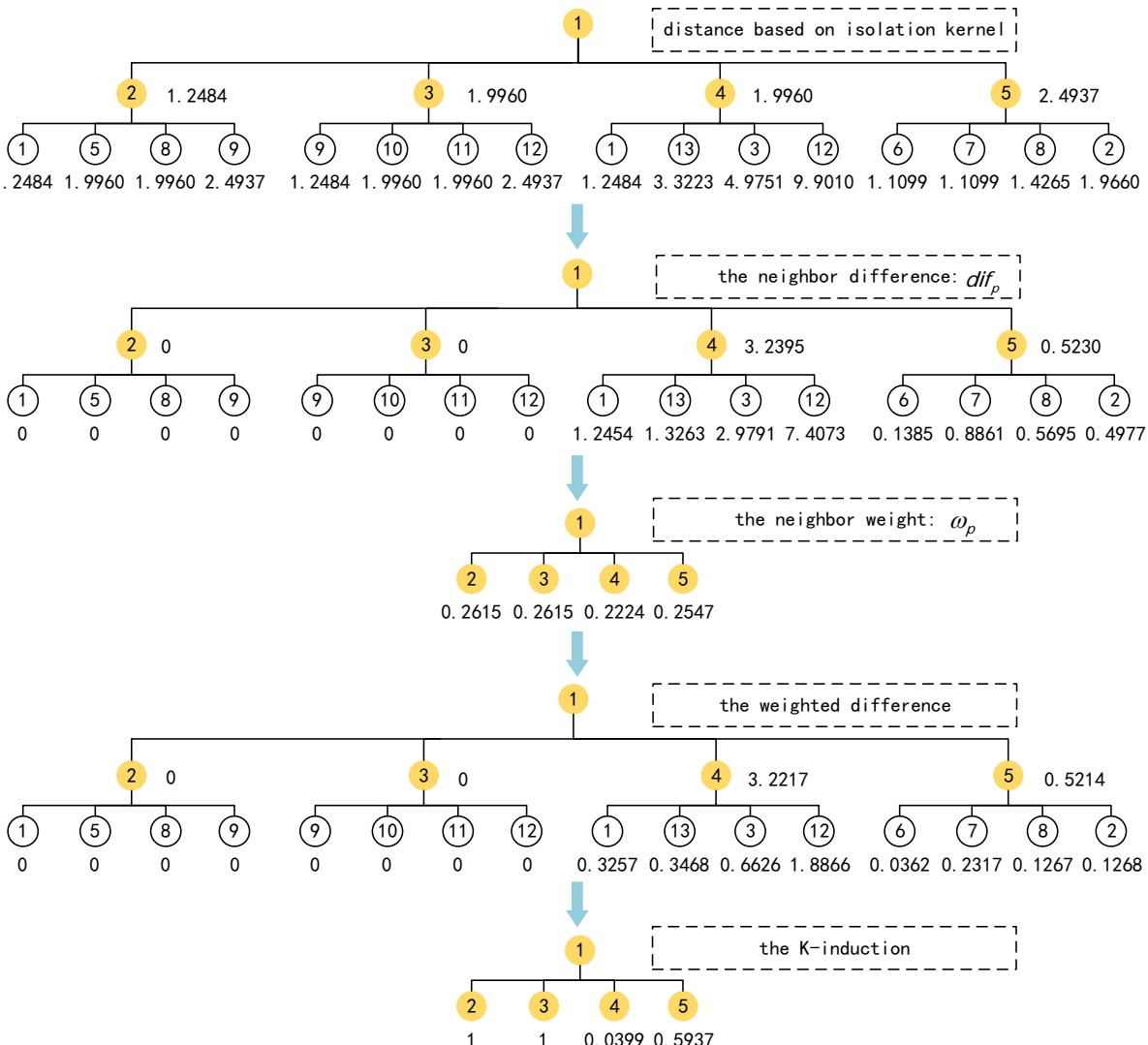

**Figure 2.** The detailed steps of K-induction similarity.

## 4. Experiment Analysis

To validate IKDC, we carried out experiments on 10 synthetic datasets and 8 real datasets. These datasets for this paper are from http://archive.ics.uci.edu/ml/index.php, accessed on 29 November 2022.

The details of the datasets are shown in Tables 2 and 3. This paper uses Accuracy (ACC), Normalized Mutual Information (NMI), and Adjusted Rand Index (ARI) [53] to evaluate the performance of the algorithm. The maximum value of the above three indicators is 1. The closer the value of the indicators is to 1, the stronger the clustering performance.

To reflect the experimental results of the algorithm more objectively, in this paper, the variable parameters of different datasets are set within the allowable range, and the most suitable parameters are obtained by repeating the experiment many times. The experiment is repeated 100 times independently according to this method, and the average results are recorded.

**Table 2.** Synthetic datasets.

| Dataset | Instances | Dimensions | Clusters |
|---|---|---|---|
| Aggregation | 788 | 2 | 7 |
| Flame | 240 | 2 | 2 |
| Jain | 373 | 2 | 2 |
| Pathbased | 300 | 2 | 3 |
| R15 | 600 | 2 | 15 |
| Spiral | 312 | 2 | 3 |
| D31 | 3100 | 2 | 31 |
| DIM512 | 1024 | 512 | 16 |
| S2 | 5000 | 2 | 15 |
| Compound | 399 | 2 | 6 |

**Table 3.** Real datasets.

| Dataset | Instances | Dimensions | Clusters |
|---|---|---|---|
| Wine | 178 | 13 | 3 |
| WDBC | 569 | 30 | 2 |
| Seeds | 210 | 7 | 3 |
| Libras | 360 | 90 | 15 |
| Ionosphere | 351 | 34 | 2 |
| Waveform | 5000 | 21 | 3 |
| Waveform (noise) | 5000 | 40 | 3 |
| Spectrometer | 531 | 102 | 48 |

*4.1. Experiments on Synthetic Datasets*

In this subsection, the effectiveness of IKDC on the 10 synthetic datasets in Table 2 will be demonstrated. For ACC, NMI, and ARI, IKDC is compared with DPC [25], DB-SCAN [24], KNN-DPC [28], SNN-DPC [34], and FCM [54]. The experimental results of the six algorithms on different datasets are presented in Tables 4 and 5. Figures 3–7 can more intensively display the experimental results of the six different algorithms on the Aggregation, Flame, and Jain datasets, where different clusters are distinguished by color, In Figure 7, the solid dots of the same color represent the core domain, and the dots of the same color and different shapes represent the boundary domain of this cluster. Since DIM512 is a high-dimensional dataset, no graph is formed to show its clustering results in this paper.

As shown in Figures 3a, 4a, 5a, 6a and 7a, the clustering effect of IKDC is improved over the other algorithms on the Aggregation dataset, with DBSCAN having the worst effect. IKDC was 0.024, 0.034, and 0.038 higher than DBSCAN in ACC, NMI, and ARI values, respectively. As shown in Figures 3b, 4b, 5b, 6b and 7b, for the Flame dataset, IKDC, DPC, DBSCAN, and KNN-DPC all perform well, and FCM has the worst clustering result. As shown in Figures 3c, 4c, 5c, 6c and 7c, for dataset Jain, the clustering results of both IKDC and SNN-DPC are superior. As shown in Table 4, on the Spiral dataset, IKDC, DPC, DBSCAN, KNN-DPC, and SNN-DPC all have good clustering effects, and only FCM has the worst clustering effect. Compared with FCM, IKDC improved the ACC, NMI, and ARI values by 0.66, 1, and 1.006, respectively. This shows that after introducing the fuzzy domain, IKDC is more effective.

From Table 5, it can be concluded that IKDC outperforms the other five algorithms in clustering on the DIM512 dataset with higher dimensionality. In comparison with FCM with the worst clustering effect, the ACC, NMI, and ARI values of IKDC are improved by 0.223, 0.296, and 0.314. Therefore, IKDC has a better clustering effect in high-dimensional space.

From Tables 4 and 5, it can be concluded that IKDC outperforms the other five clustering algorithms for the remaining five datasets, especially when the datasets are unevenly distributed and of high dimensionality.

To sum up, by comprehensively considering the performance of the six algorithms in Tables 4 and 5 on the three clustering indicators and Figures 3–8, it can be concluded that IKDC outperforms the other five algorithms on the 10 synthetic datasets, which verifies that IKDC has a superior clustering effect.

**Table 4.** ACC, NMI, and ARI of six algorithms on different synthetic datasets.

| Algorithm | ACC | NMI | ARI | Algorithm | ACC | NMI | ARI |
|---|---|---|---|---|---|---|---|
| Aggregation | | | | Spiral | | | |
| IKDC | **1.000** | **1.000** | **1.000** | IKDC | **1.000** | **1.000** | **1.000** |
| DPC | 0.995 | 0.992 | 0.990 | DPC | **1.000** | **1.000** | **1.000** |
| DBSCAN | 0.973 | 0.958 | 0.958 | DBSCAN | **1.000** | **1.000** | **1.000** |
| KNN-DPC | 0.997 | 0.992 | 0.996 | KNN-DPC | **1.000** | **1.000** | **1.000** |
| SNN-DPC | 0.978 | 0.955 | 0.959 | SNN-DPC | **1.000** | **1.000** | **1.000** |
| FCM | 0.778 | 0.825 | 0.684 | FCM | 0.340 | 0.000 | -0.006 |
| Flame | | | | D31 | | | |
| IKDC | **1.000** | **1.000** | **1.000** | IKDC | 0.970 | 0.962 | 0.953 |
| DPC | **1.000** | **1.000** | **1.000** | DPC | 0.968 | 0.958 | 0.936 |
| DBSCAN | **1.000** | **1.000** | **1.000** | DBSCAN | 0.968 | 0.957 | 0.935 |
| KNN-DPC | **1.000** | **1.000** | **1.000** | KNN-DPC | 0.970 | 0.960 | 0.940 |
| SNN-DPC | 0.998 | 0.899 | 0.950 | SNN-DPC | **0.974** | **0.963** | **0.974** |
| FCM | 0.850 | 0.442 | 0.488 | FCM | 0.891 | 0.862 | 0.936 |
| Jain | | | | DIM512 | | | |
| IKDC | **1.000** | **1.000** | **1.000** | IKDC | **0.966** | **0.951** | **0.956** |
| DPC | 0.981 | 0.976 | 0.970 | DPC | 0.944 | 0.940 | 0.935 |
| DBSCAN | 0.928 | 0.895 | 0.890 | DBSCAN | 0.851 | 0.774 | 0.749 |
| KNN-DPC | 0.970 | 0.960 | 0.940 | KNN-DPC | 0.918 | 0.897 | 0.890 |
| SNN-DPC | **1.000** | **1.000** | **1.000** | SNN-DPC | 0.939 | 0.896 | 0.926 |
| FCM | 0.778 | 0.831 | 0.707 | FCM | 0.743 | 0.655 | 0.6428 |

**Table 5.** ACC, NMI, and ARI of six algorithms on different synthetic datasets.

| Algorithm | ACC | NMI | ARI | Algorithm | ACC | NMI | ARI |
|---|---|---|---|---|---|---|---|
| Pathbased | | | | S2 | | | |
| IKDC | **0.980** | **0.920** | **0.916** | IKDC | **0.966** | **0.951** | **0.956** |
| DPC | 0.753 | 0.555 | 0.472 | DPC | 0.944 | 0.940 | 0.935 |
| DBSCAN | 0.823 | 0.731 | 0.613 | DBSCAN | 0.851 | 0.774 | 0.749 |
| KNN-DPC | 0.760 | 0.561 | 0.561 | KNN-DPC | 0.918 | 0.897 | 0.890 |
| SNN-DPC | 0.977 | 0.901 | 0.929 | SNN-DPC | 0.939 | 0.896 | 0.926 |
| FCM | 0.747 | 0.550 | 0.465 | FCM | 0.741 | 0.690 | 0.694 |
| R15 | | | | Compound | | | |
| IKDC | **1.000** | **1.000** | **1.000** | IKDC | **0.885** | **0.913** | **0.899** |
| DPC | 0.997 | 0.994 | 0.993 | DPC | 0.832 | 0.873 | 0.833 |
| DBSCAN | 0.993 | 0.989 | 0.986 | DBSCAN | 0.840 | 0.828 | 0.844 |
| KNN-DPC | 0.997 | 0.994 | 0.993 | KNN-DPC | 0.870 | 0.552 | 0.809 |
| SNN-DPC | 0.997 | 0.994 | 0.993 | SNN-DPC | 0.857 | 0.853 | 0.835 |
| FCM | 0.997 | 0.965 | 0.993 | FCM | 0.501 | 0.619 | 0.406 |

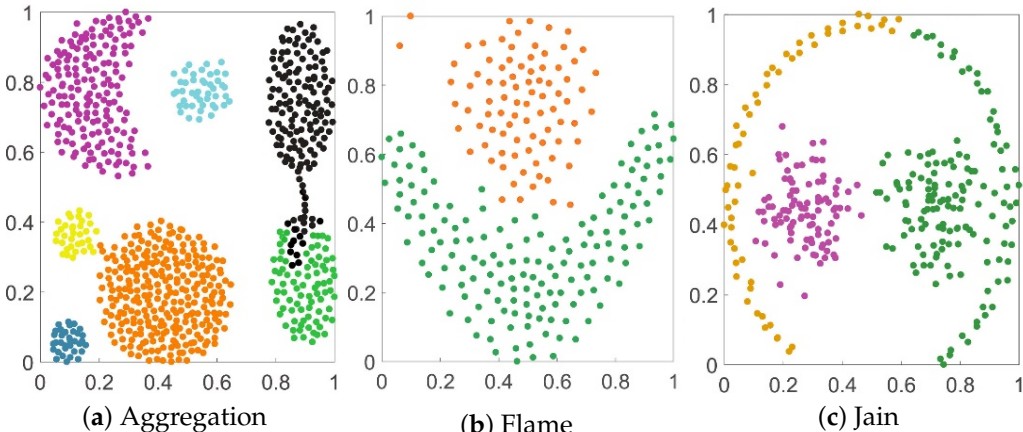

**Figure 3.** Clustering effect of the DBSCAN algorithm on Aggregation, Flame, and Jain.

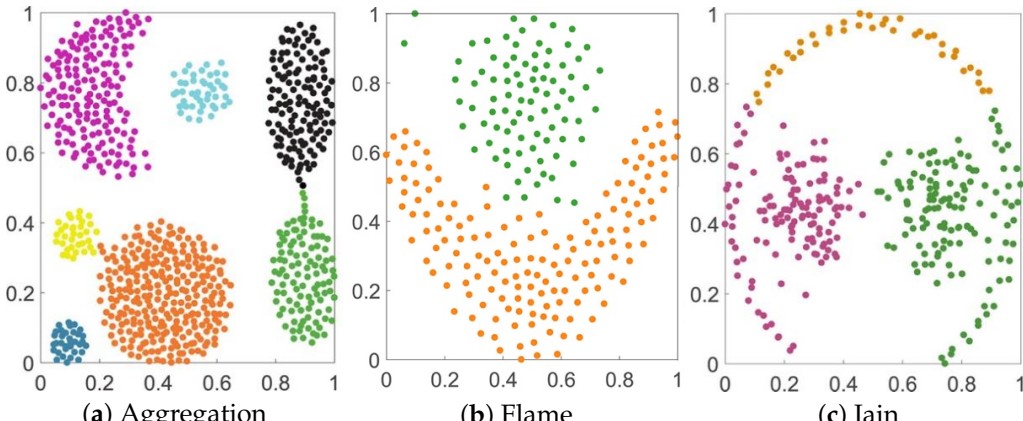

**Figure 4.** Clustering effect of the KNN-DPC algorithm on Aggregation, Flame and Jain.

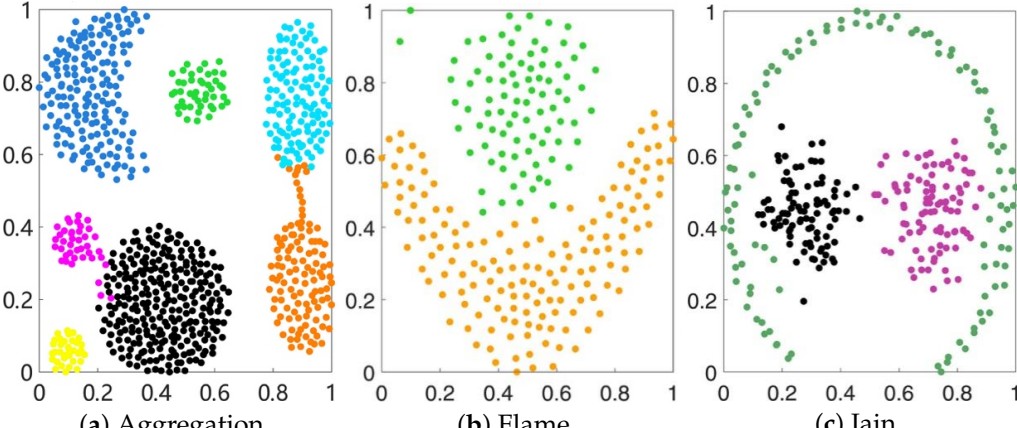

**Figure 5.** Clustering effect of the SNN-DPC algorithm on Aggregation, Flame, and Jain.

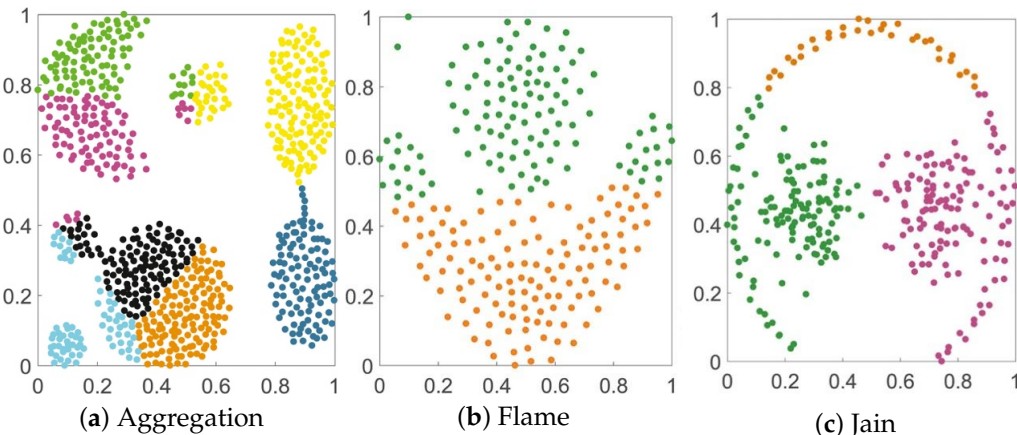

**Figure 6.** Clustering effect of the FCM algorithm on Aggregation, Flame, and Jain.

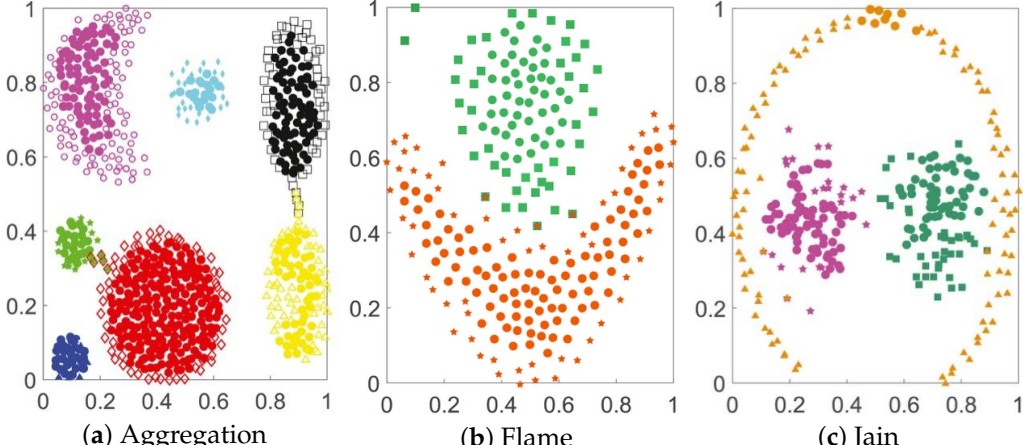

**Figure 7.** Clustering effect of the IKDC algorithm on Aggregation, Flame, and Jain.

### 4.2. Experiments on Real Datasets

In this subsection, the validity of IKDC on the eight real datasets in Table 3 will be shown. For ACC, NMI, and ARI, IKDC is compared with DPC, DBSCAN, KNN-DPC, SNN-DPC, and FCM. Tables 6 and 7 show the comparison results of the six algorithms.

As shown in Tables 6 and 7, IKDC has markedly more clustering effects for the Ionosphere, Wine, Waveform, Waveform (noise), Libras, WDBC, Spectrometer, and Seeds datasets. On Ionosphere and WDBC datasets, IKDC has improved values in ACC, NMI, and ARI compared to SNN-DPC and FCM. Overall, the clustering performance of IKDC outperforms the other five algorithms in high-dimensional space. On the Libras dataset, although the clustering effect of IKDC is the same as that of DPC, IKDC introduces three-way clustering, which is more consistent with human cognition, and IKDC outperforms the other five algorithms. On the other datasets, IKDC significantly outperforms the other five algorithms.

To sum up, by comprehensively considering the performance of the six algorithms on the three indicators, it is demonstrated that the IKDC outperforms the other five clustering algorithms on the eight real datasets, which verifies that the clustering effect of IKDC is more excellent.

**Table 6.** ACC, NMI, and ARI of six algorithms on different real datasets.

| Algorithm | ACC | NMI | ARI | Algorithm | ACC | NMI | ARI |
| --- | --- | --- | --- | --- | --- | --- | --- |
| Ionosphere | | | | Waveform | | | |
| IKDC | **1.000** | **1.000** | **1.000** | IKDC | **0.980** | **0.984** | **0.955** |
| DPC | **1.000** | **1.000** | **1.000** | DPC | 0.968 | 0.958 | 0.936 |
| DBSCAN | **1.000** | **1.000** | **1.000** | DBSCAN | 0.524 | 0.152 | 0.135 |
| KNN-DPC | **1.000** | **1.000** | **1.000** | KNN-DPC | 0.635 | 0.218 | 0.223 |
| SNN-DPC | 0.998 | 0.899 | 0.950 | SNN-DPC | 0.598 | 0.326 | 0.311 |
| FCM | 0.850 | 0.442 | 0.488 | FCM | 0.628 | 0.374 | 0.353 |
| Wine | | | | Waveform (noise) | | | |
| IKDC | 0.952 | 0.873 | 0.852 | IKDC | **0.983** | **0.979** | 0.959 |
| DPC | 0.882 | 0.710 | 0.627 | DPC | 0.969 | 0.958 | 0.936 |
| DBSCAN | 0.910 | 0.753 | 0.741 | DBSCAN | 0.968 | 0.957 | 0.935 |
| KNN-DPC | 0.904 | 0.743 | 0.727 | KNN-DPC | 0.970 | 0.960 | 0.940 |
| SNN-DPC | **0.966** | **0.878** | **0.899** | SNN-DPC | 0.974 | 0.963 | **0.974** |
| FCM | 0.949 | 0.850 | 0.834 | FCM | 0.891 | 0.862 | 0.936 |

**Table 7.** ACC, NMI, and ARI of different clustering algorithms on different real datasets.

| Algorithm | ACC | NMI | ARI | Algorithm | ACC | NMI | ARI |
| --- | --- | --- | --- | --- | --- | --- | --- |
| Libras | | | | Spectrometer | | | |
| IKDC | **0.503** | 0.543 | **0.401** | IKDC | **0.979** | **0.975** | **0.975** |
| DPC | 0.420 | 0.514 | 0.345 | DPC | 0.968 | 0.958 | 0.936 |
| DBSCAN | 0.458 | 0.626 | 0.309 | DBSCAN | 0.968 | 0.957 | 0.935 |
| KNN-DPC | 0.497 | 0.634 | 0.361 | KNN-DPC | 0.970 | 0.960 | 0.940 |
| SNN-DPC | 0.494 | **0.661** | 0.393 | SNN-DPC | 0.974 | 0.963 | 0.974 |
| FCM | 0.183 | 0.216 | 0.069 | FCM | 0.891 | 0.862 | 0.936 |
| WDBC | | | | Seeds | | | |
| IKDC | 0.917 | 0.720 | **0.751** | IKDC | **0.934** | **0.763** | **0.830** |
| DPC | 0.830 | 0.373 | 0.373 | DPC | 0.910 | 0.716 | 0.742 |
| DBSCAN | 0.631 | 0.530 | 0.530 | DBSCAN | 0.624 | 0.423 | 0.487 |
| KNN-DPC | 0.870 | **0.766** | 0.436 | KNN-DPC | 0.914 | 0.734 | 0.766 |
| SNN-DPC | 0.874 | 0.535 | 0.650 | SNN-DPC | 0.924 | 0.754 | 0.789 |
| FCM | **0.928** | 0.609 | 0.631 | FCM | 0.900 | 0.691 | 0.727 |

*4.3. The Influence of Own Parameters on the Experimental Results*

In this subsection, to verify the effect of its own parameters on the clustering effect, we conducted experiments with the Wine dataset as an example, and the experimental results of the control parameter $\psi$ and parameter $t$ are shown in Figure 8. When $\psi = \text{len(wine)}/5$, as the number of isolation trees increases, the number of wrong points becomes less and less. When the number of isolation trees is the same, as $\psi$ increases, the experimental results are better. Therefore, the experiments show that when the number of $\psi$ and $t$ is within the maximum limit, the clustering effect is better.

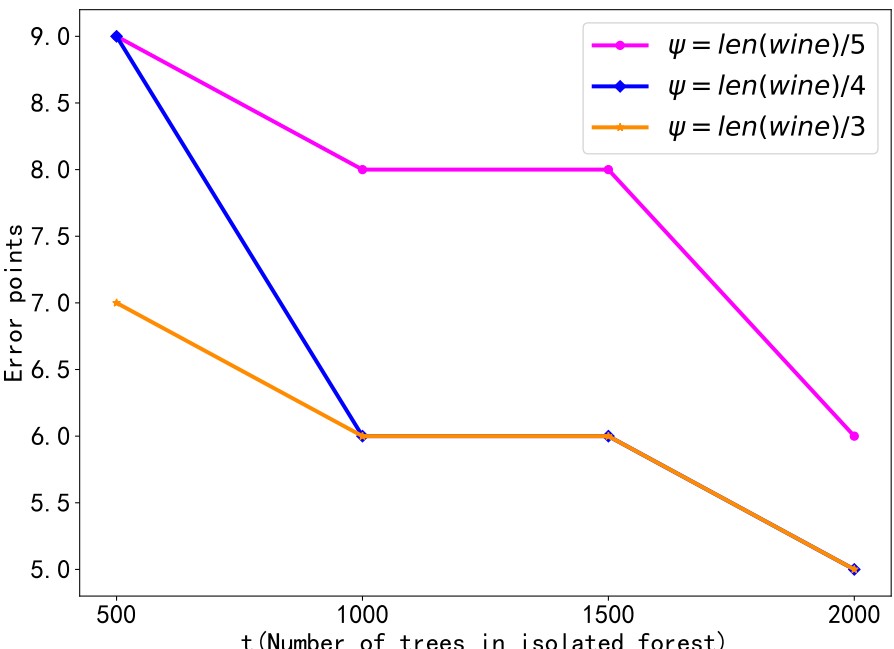

**Figure 8.** Influence of own parameters on experimental results.

## 5. Discussion

The limitations of this paper mainly lie in two points: (1) the need to manually set parameter *K* in the nearest neighbor; and (2) the calculation of K-induction is more complicated. To ameliorate the above limitations, in the future, we can proceed in the following two orientations: (1) finding the most suitable *K* through an adaptive method; and (2) a deep neural network can be utilized to assign the leftover points.

## 6. Conclusions

The proposed IKDC makes two improvements to the DPC. The first improvement is to introduce an isolation kernel to replace the distance calculation metric in the DPC, which effectively solves the "curse of dimensionality" problem and avoids the failure of traditional distance formulas for high-dimensional samples. The second improvement is to use the K-induction to assign the leftover points, which can effectively avoid the transmission of the error message and prevent the domino effect. Through experiments on several high-dimensional datasets, this paper compares the clustering performance of the IKDC with five representative algorithms (including DPC, DBSCAN, KNN-DPC, SNN-DPC, and FCM) for ACC, NMI, and ARI. The IKDC algorithm works better for high-dimensional samples obtained by comparing the experimental results.

**Author Contributions:** Conceptualization, S.Z. and K.L.; methodology, S.Z.; software, S.Z.; validation, S.Z. and K.L. ; formal analysis, S.Z.; investigation, S.Z.; resources, S.Z.; data curation, S.Z.; writing—original draft preparation, S.Z.; writing—review and editing, S.Z. and K.L.; visualization, S.Z. and K.L.; supervision, K.L. All authors have read and agreed to the published version of the manuscript.

**Funding:** This research received no external funding.

**Data Availability Statement:** The data used to support the findings of this study are included at http://archive.ics.uci.edu/ml/index.php, accessed on 29 November 2022.

**Conflicts of Interest:** The authors declare no conflict of interest.

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
