# Peer review of "A Novel Density Peaks Clustering Algorithm with Isolation Kernel and K-Induction"

_applsci, doi:10.3390/app13010322_

Round 1

Reviewer 1 Report

The topic is of some interests, and the result looks reasonable based on the method presented.

Some english and format of the paper need to be revised. For example, "Where" after equations should be read as "where".

Author Response

Comment :

  1. Some english and format of the paper need to be revised. For example, "Where" after equations should be read as "where".

Response:

Thanks for your careful check. Based on your comments, we changed all "Where" after the equation to "where" and corrected other errors in text formatting and grammar. All corrections have been marked with the "Track Changes" function in the revised manuscript.

Reviewer 2 Report

This paper proposes a novel density peaks clustering algorithm with isolation kernel and K-induction. The novel algorithm can surmount the deficiencies of the original DPC algorithm of producing bad labels. Overall, the methodology is well-presented, and the manuscript is well-written. It can be accepted after some minor revisions: The caption of Figs. 3-7 should be more specific, i.e., the algorithm should be included.

Author Response

Comment 1:

  1. The caption of Figs. 3-7 should be more specific, i.e., the algorithm name should be included.

Response:

We sincerely thank the reviewer for careful reading. As suggested by the reviewer, we have corrected the caption of Figs. 3-7 to specify the names of the algorithms and datasets, making the titles more specific.

Comment 2:

  1. The aesthetic of the figures and the equations within the manuscript can be further improved. Specifically, the size of the text in the figures should be uniform; the authors should check whether some variables within the equations and the figures should be italic.

Response:

Based on your comments, we have unified the size and format of the text in the figures, changed all the variables in the equations and the manuscript to italics, and corrected wrong equations (Such as Equation.3) and wrong variables in the manuscript. At the same time, we have corrected Figure. 1, 2, and 8 to make the numbers in the pictures more obvious, thereby improving the beauty of the figures.

Comment 3:

  1. (Optional) The hyperparameter dc is an important input of the DPC and the proposed IKDC algorithm. The manuscript and the proposed algorithm would be better improved if it can be selected self-adaptively.

Response:

As you are concerned, we are also trying to solve the problem of parameter adaptation. We will tackle the problem of parameter adaptation in the next paper.

Reviewer 3 Report

This paper proposes a density peaks clustering algorithm with an isolation kernel and K-induction named IKDC It uses an isolation kernel to replace the distance calculation metric in the density peak clustering that can solve the curse of dimensionality problem and avoids the failure of traditional distance formulas for high-dimensional samples. It also uses the K-induction to assign the leftover points, which can effectively avoid the transmission of the error message and prevent the domino effect. The experiment was conducted on both synthetic and real datasets to compare the performance of the proposed algorithm with the other five algorithms using some external evaluation metrics.

In general, the paper is well-written and well-organized. The topic is interesting and suitable for the scope of the journal. The experimental results seem to be reasonable. In support of this paper, the authors should consider the following points to further improve the paper's quality.

- In the Abstract, avoid using mathematical notations of core and boundary domains that may hamper the willingness to continue reading the paper of potential readers.

- The Introduction should give some info on several other clustering methods. I have used k-means, DBSCAN, and HDBSCAN- an improved version of DBSCAN that allows varying density clusters instead of using a global epsilon distance as in DBSCAN. I observed that k-means, DBSCAN, and HDBSCAN could perform very well in the clustering task. In some cases, DBSCAN and HDBSCAN were even better than k-means since they can remove noises. I also used the HDBSCAN python version. It is an efficient algorithm in terms of runtime and can work well with high-dimensional data. Thus, the authors should also summarize/compare the strength and weaknesses of HDBSCAN [https://hdbscan.readthedocs.io/en/latest/comparing_clustering_algorithms.html]. In addition, discuss possible methods that can perform clustering for an unknown number of clusters and provide high interpretability, such as hierarchical clustering, the authors can refer [https://doi.org/10.1007/978-981-15-1209-4_1] in the discussion.  From that, highlight the main advantages of the proposed method over other methods.

- In section 2, put a table of main notations used in the paper.

- In section 3, theoretically discuss the completeness of IKDC.

- The authors should use boldface fonts to highlight the best performance for each evaluation metric in Tables 3-6.

- Proofread the paper to fix all typos. There are many places in the paper the authors put no white space before "(" or "[". For instance, data analysis tool [1,2 ] instead of data analysis tool[1,2 ]; Definition 1 (distance based on isolation kernel) instead of Definition1(distance based on isolation kernel)

I donot want to see this problem in the revised version!

Author Response

Comment 1:

  1. In the Abstract, avoid using mathematical notations of core and boundary domains that may hamper the willingness to continue reading the paper of potential readers.

Response:

We feel great thanks for your professional review work on our manuscript. Based on your comments, we have removed mathematical notation in the abstract for core and boundary domains that may hinder a potential reader's willingness to continue reading the paper (See lines 9 and 10 on page 1 for details).

Comment 2:

  1. The Introduction should give some info on several other clustering methods. I have used k-means, DBSCAN, and HDBSCAN- an improved version of DBSCAN that allows varying density clusters instead of using a global epsilon distance as in DBSCAN. I observed that k-means, DBSCAN, and HDBSCAN could perform very well in the clustering task. In some cases, DBSCAN and HDBSCAN were even better than k-means since they can remove noises. I also used the HDBSCAN python version. It is an efficient algorithm in terms of runtime and can work well with high-dimensional data. Thus, the authors should also summarize/compare the strength and weaknesses of HDBSCAN [https://hdbscan.readthedocs.io/en/latest/comparing_clustering_algorithms.html]. In addition, discuss possible methods that can perform clustering for an unknown number of clusters and provide high interpretability, such as hierarchical clustering, the authors can refer [https://doi.org/10.1007/978-981-15-1209-4_1] in the discussion. From that, highlight the main advantages of the proposed method over other methods.

Response:

We have added the discussion of k-means, DBSCAN, HDBSCAN, and how to determine the optimal number of clusters to the Introduction section of the manuscript. Their advantages and disadvantages are analyzed, and compared with the proposed method, the superiority of the proposed method is proved theoretically. At the same time, we have added relevant content to the literature (See literature [23] and [26] for details).

The specific location to add discussion is as follows:

1) Lines 24-31and 35-38 on page 1;

2) Lines 38-44 on page 2;

3) Lines 99-100 on page 3;

Comment 3:

  1. In section 2, put a table of main notations used in the paper.

Response:

Based on your comments, we have added a table of the main notations used in the paper at the beginning of the second section (See line 116 on page 3 for details).

Comment 4:

  1. In section 3, theoretically discuss the completeness of IKDC.

Response:

Following your suggestion, the discussion of the completeness of the proposed method has been added to the third subsection of our manuscript at the following locations:

1) Lines 163-168 and 184-187 on page 5;

2) Lines 195-198 and 207-208 on page 6;

3) Lines 209-210 on page 7.

Comment 5:

  1. The authors should use boldface fonts to highlight the best performance for each evaluation metric in Tables 3-6.

Response:

As suggested by the reviewer, we have used boldface fonts to highlight the best performance for each evaluation metric in Tables 3-6 (See pages 11, 13 and 14 for details).

Comment 6:

  1. Proofread the paper to fix all typos. There are many places in the paper the authors put no white space before "(" or "[". For instance, data analysis tool [1,2 ] instead of data analysis tool[1,2 ]; Definition 1 (distance based on isolation kernel) instead of Definition1(distance based on isolation kernel)

Response:

Based on your comments, we have carefully checked every "[" and "(" in the manuscript, added a space in front of each "[" and "(", and corrected other formatting errors in the manuscript. 

Round 2

Reviewer 3 Report

I have checked the revisions. I am very happy that the authors have carefully solved all of my comments. The quality of the paper has been improved significantly. I will vote for an acceptance.